# Prevalence and correlates of prescription opioid use among US adults, 2019–2020

**Anna Zajacova**[1]*, **Hanna Grol-Prokopczyk**[2], **Merita Limani**[1], **Christopher Schwarz**[3], **Ian Gilron**[4]

1 Department of Sociology, University of Western Ontario, London, Ontario, Canada, 2 Department of Sociology, University at Buffalo, State University of New York, Buffalo, New York, United States of America, 3 Department of Politics, New York University, New York, New York, United States of America, 4 Department of Anesthesiology and Perioperative Medicine, Queen's University School of Medicine, Kingston, Ontario, Canada

* anna.zajacova@uwo.ca

**Data Availability Statement:** The data are publicly available for download at https://nhis.ipums.org/nhis/.

## Abstract

This study estimates the prevalence of prescription opioid use (POU) in the United States (US) in 2019–2020, both in the general population and specifically among adults with pain. It also identifies key geographic, demographic, and socioeconomic correlates of POU. Data were from the nationally-representative National Health Interview Survey 2019 and 2020 (N = 52,617). We estimated POU prevalence in the prior 12 months among all adults (18+), adults with chronic pain (CP), and adults with high-impact chronic pain (HICP). Modified Poisson regression models estimated POU patterns across covariates. We found POU prevalence of 11.9% (95% CI 11.5, 12.3) in the general population, 29.3% (95% CI 28.2, 30.4) among those with CP, and 41.2% (95% CI 39.2, 43.2) among those with HICP. Findings from fully-adjusted models include the following: In the general population, POU prevalence declined about 9% from 2019 to 2020 (PR = 0.91, 95% CI 0.85, 0.96). POU varied substantially across US geographic regions: It was significantly more common in the Midwest, West, and especially the South, where adults had 40% higher POU (PR = 1.40, 95% CI 1.26, 1.55) than in the Northeast. In contrast, there were no differences by rural/urban residence. In terms of individual characteristics, POU was lowest among immigrants and among the uninsured, and was highest among adults who were food insecure and/or not employed. These findings suggest that prescription opioid use remains high among American adults, especially those with pain. Geographic patterns suggest systemic differences in therapeutic regimes across regions but not rurality, while patterns across social characteristics highlight the complex, opposing effects of limited access to care and socioeconomic precarity. Against the backdrop of continuing debates about benefits and risks of opioid analgesics, this study identifies and invites further research about geographic regions and social groups with particularly high or low prescription opioid use.

**Funding:** Research reported in this analysis was supported by the National Institute on Aging of the National Institutes of Health under Award Number R01AG065351 (PI: Grol-Prokopczyk) and by the Social Sciences and Humanities Research Council of Canada (SSHRC) Insight Grant (PI: Zajacova). The content is solely the responsibility of the authors and does not necessarily represent the official views of the National Institutes of Health or SSHRC. The funders had no role in study design, data collection and analysis, decision to publish, or preparation of the manuscript.

**Competing interests:** The authors have declared that no competing interests exist.

## Introduction

The US has a crisis of pain [1] and its undertreatment [2], but also a parallel crisis of overreliance on opioids and related opioid misuse or illicit use, overdose, and mortality [3,4]. Given the close links of *prescription* opioid use (POU) with all these phenomena, it is important to characterize and understand geographic and social patterns in POU. While there is a vast literature on opioid use, surprisingly little describes up-to-date patterns in POU among US adults. In particular, prior research leaves unclear how POU varies across United States (US) regions and rural/urban areas. More is known about demographic and socioeconomic patterns, but many open questions remain (as discussed just below). Four aspects of existing research present barriers to a full understanding of current patterns in POU in the US.

First, many recently published studies on POU among US adults are based on data over a decade old [5–8]. Given the dramatic changes in opioid prescribing in recent years [9] alongside increases in pain prevalence [1], information from a decade or longer ago may not reflect current POU patterns. Second, studies often utilize data that has only limited information on respondent characteristics [10], especially in pharmacy prescription-based studies [11]. Alternatively, they focus on a single predictor at a time [6]; and/or examine only bivariate associations [12]. This is problematic because multiple social determinants of POU are correlated, and omitting some will bias the estimates for others. For instance, rural areas have older populations with more white and US-born adults [13], lower socioeconomic status (SES) [14], and more pain [13], characteristics also associated with higher POU [9,15,16]. Thus, if not taking these correlated characteristics into account, estimates for rural/urban differences could be biased by the effects of age, race and ethnicity, SES, and/or pain differences. Third, nearly all research has focused either on the general population or adults who are treated for pain, but not both. Adults who receive treatment for pain are a select group—not equivalent to all adults with pain—since many adults with pain do not receive medical treatment for pain. And POU in the general population, if not conditioning on pain, may reflect pain disparities *or* POU disparities; most studies do not clarify which. Fourth, some studies use opioid dispensing information at county or state levels or other aggregate approaches, raising the risk of ecological fallacy. For instance, counties with higher proportions of uninsured adults have higher opioid dispensing amounts [9]. However, the county-level proportion of uninsured captures the county's socioeconomic status, rather than any given individual's access to medical care and thus to opioid prescriptions; at the individual level, the association between health insurance and opioid dosage may differ from that observed at the county level.

Our analysis overcomes these four issues. It presents the most comprehensive portrait of prescription opioid use in the US based on large, up-to-date, nationally-representative, individual-level self-reported data. We use National Health Interview Survey (NHIS) data from 2019 (when questions about opioid use were first included) and 2020 (the most recent data available at time of writing). We consider the impact of geographic, demographic, socioeconomic, and health insurance correlates together. We examine POU in three subsamples: the general population, adults with chronic pain, and adults with high-impact chronic pain. By presenting POU patterns in the total population and then conditioning the estimates on pain prevalence and interference, we are able to isolate POU disparities from underlying pain disparities.

## Methods

### Data

Our data come from the 2019 and 2020 National Health Interview Survey (NHIS) harmonized by IPUMS at the University of Minnesota [17]. The NHIS is the leading source of information

about health in the US. The sample is representative of the non-institutionalized civilian population. The raw data, harmonized for pooling multiple waves by IPUMS, are available at https://nhis.ipums.org/nhis/. Response rates were 59.1% in 2019 and 48.9% in 2020. Detailed information on response rates, sample sizes, and other key features of the survey are available online for 2019 [18] and 2020 [19]. We include adults 18 years and older in the analysis. Analytic sample sizes are summarized in the Results section as per STROBE guidelines. This analysis uses only de-identified and publicly available data; it was therefore exempt from IRB review.

## Variables

The outcome is a dichotomous indicator for taking prescription opioids during the past 12 months. First, all respondents were asked "At any time in the past 12 months, did you take prescription medication?" Those who answered yes were then asked, "During the past 12 months, have you taken any opioid pain relievers prescribed by a doctor, dentist, or other health professional?" The term 'opioid' is explained in the survey, with examples. Full details about the prompt are available online [20, p.287].

**Pain experience.**   We examined prescription opioid use in three samples: 1) all adults, regardless of pain status, to capture POU in the general population; 2) adults who reported they had chronic pain (CP), defined as pain on most days or every day (versus never or some days) in the last 3 months; and 3) adults with high-impact chronic pain (HICP), defined as pain that limited life or work activities on most days or every day (versus never or some days) over the past 3 months. These pain definitions are well established in prior epidemiological work [21–24].

**Covariates.**   Year of interview is 2020 versus 2019 as reference. Geographic indicators are region of residence (Northeast as reference, Midwest, South, and West) and rural/urban designation (large central metropolitan area as reference, large fringe metro, medium and small metro, and nonmetropolitan). Demographic characteristics are age (18–44 as reference, 45–64, and 65+), dichotomous gender with male as reference, racial/ethnic self-identification (non-Hispanic white as reference, non-Hispanic Black, Hispanic, and 'other,'), immigrant status (US-born as reference versus foreign-born), and marital status (married as reference, previously married, and never married). Socioeconomic status is measured with five variables. Educational attainment is categorized as less than high school, high school, some college, associate degree, bachelor's degree, and master's degree or higher as reference. Household income categories are used in the full detail provided by IPUMS: $0–34,999, $35,000–49,999, $50,000-$74,999, $75,000–99,999, and $100,000+ as reference. Our food insecurity indicator is based on a ten-question scale, administered by NHIS following the US Department of Agriculture Guide, which captures a family's "access by all people at all times to enough food for an active, healthy life" [25, p.6]. The battery includes questions such as whether anyone in the family ate less or skipped meals because of not enough money, or ever went hungry, or whether the family couldn't afford to eat balanced meals. All questions were asked with respect to the prior month. We dichotomized this scale as 1 for a positive response to any food insecurity item versus 0 for food-secure as reference. Employment status, measured with respect to the prior 1 to 2 weeks, was dichotomized as employed (reference) versus not employed. Finally, a health insurance variable was created from a set of dichotomous indicators for different insurance types and classified as private insurance (reference), public insurance, and no insurance. Respondents with both public and private insurance, mostly older adults with Medicare and supplemental insurance, were coded as having private insurance.

All variables are self-reported.

## Statistical analyses

First, we estimated weighted unadjusted prevalence of POU in the three analytic subsamples (all adults, those with CP, and those with HICP). In each subsample, POU prevalence was estimated for the total, and also by age and sex. The prevalence estimates and their 95% confidence intervals are summarized in Table 1 and visualized in Fig 1. We also summarized the distribution of all covariates in the sample population and present the results in S1 Table in S1 File.

Second, we estimated robust Poisson models of POU in each of the three analytic subsamples. These models were fully-adjusted; that is, they jointly included all covariates listed above, in order to assess the role of each covariate net of other characteristics. The robust, or modified, Poisson regression is appropriate for binary dependent variables and preferred over alternatives like logistic regression or the log-binomial model due to its interpretability and robustness to certain forms of misspecification [26]. Additionally, exponentiated coefficient from the robust Poisson regression yield prevalence ratios, which are easy to understand and disseminate for experts and non-experts alike. In contrast, while logistic regression is widely used, its results when exponentiated yield odds ratios, which are non-intuitive to interpret and sometimes mistakenly interpreted as prevalence ratios [27,28]. Results—estimates and their 95% confidence intervals—from the regression models are presented using three complementary perspectives. Table 2 shows prevalence ratio estimates that highlight relative POU levels across population characteristics; the results are also visualized in Fig 2. We then recast the results into absolute terms as average predicted prevalence of POU (S2 Table in S1 File). These estimates summarize the mean prevalence of POU in each population group. Readers may be more familiar with adjusted predictions at the means, that is, predicted POU prevalence for each level of a variable holding other covariates at their means [29]. However, this approach is suboptimal in our study because our covariates are all categorical; their means thus do not correspond to any meaningful characteristic (e.g., mean sex or mean race do not have any informative interpretation). We therefore estimate average adjusted predictions, which calculate the POU prevalence associated with a given characteristic (say, being food insecure) while holding all covariate values at their actual levels.

All analyses are weighted using appropriate weights including adjustment for pooling across waves as per NHIS IPUMS guidelines [30]. The results are from complete-case analyses, an appropriate approach given the low missingness in the data (see first paragraph of Results section and S1 Table in S1 File). Sensitivity analyses also examined POU among adults with severe pain; the findings are similar to those for HICP. We tested 2019 and 2020 data separately because of the potential impact of the COVID pandemic on the 2020 estimates; the

**Table 1. Prevalence of prior-year prescription opioid use, US adults 2019–2020.**

|  | All | | With CP | | With HICP | |
|---|---|---|---|---|---|---|
|  | Prev | 95% CI | Prev | 95% CI | Prev | 95% CI |
| All adults | 11.9 | (11.5, 12.3) | 29.3 | (28.2, 30.4) | 41.2 | (39.2, 43.2) |
| Men | 10.1 | (9.6, 10.6) | 26.3 | (24.8, 27.9) | 39.3 | (36.4, 42.3) |
| Women | 13.6 | (13.0, 14.2) | 31.7 | (30.3, 33.3) | 42.5 | (39.8, 45.3) |
| Age 18–44 | 8.7 | (8.2, 9.3) | 23.8 | (21.7, 26.1) | 34.9 | (30.4, 39.6) |
| Age 45–64 | 14.0 | (13.3, 14.8) | 32.6 | (30.8, 34.4) | 45.7 | (42.7, 48.8) |
| Age 65+ | 15.4 | (14.7, 16.2) | 29.7 | (28.1, 31.4) | 39.2 | (36.3, 42.2) |

CP = chronic pain; HICP = high-impact chronic pain; Prev = prevalence; CI = confidence interval Prevalence estimate and their 95% confidence intervals. Estimation adjusts for NHIS complex sampling design.

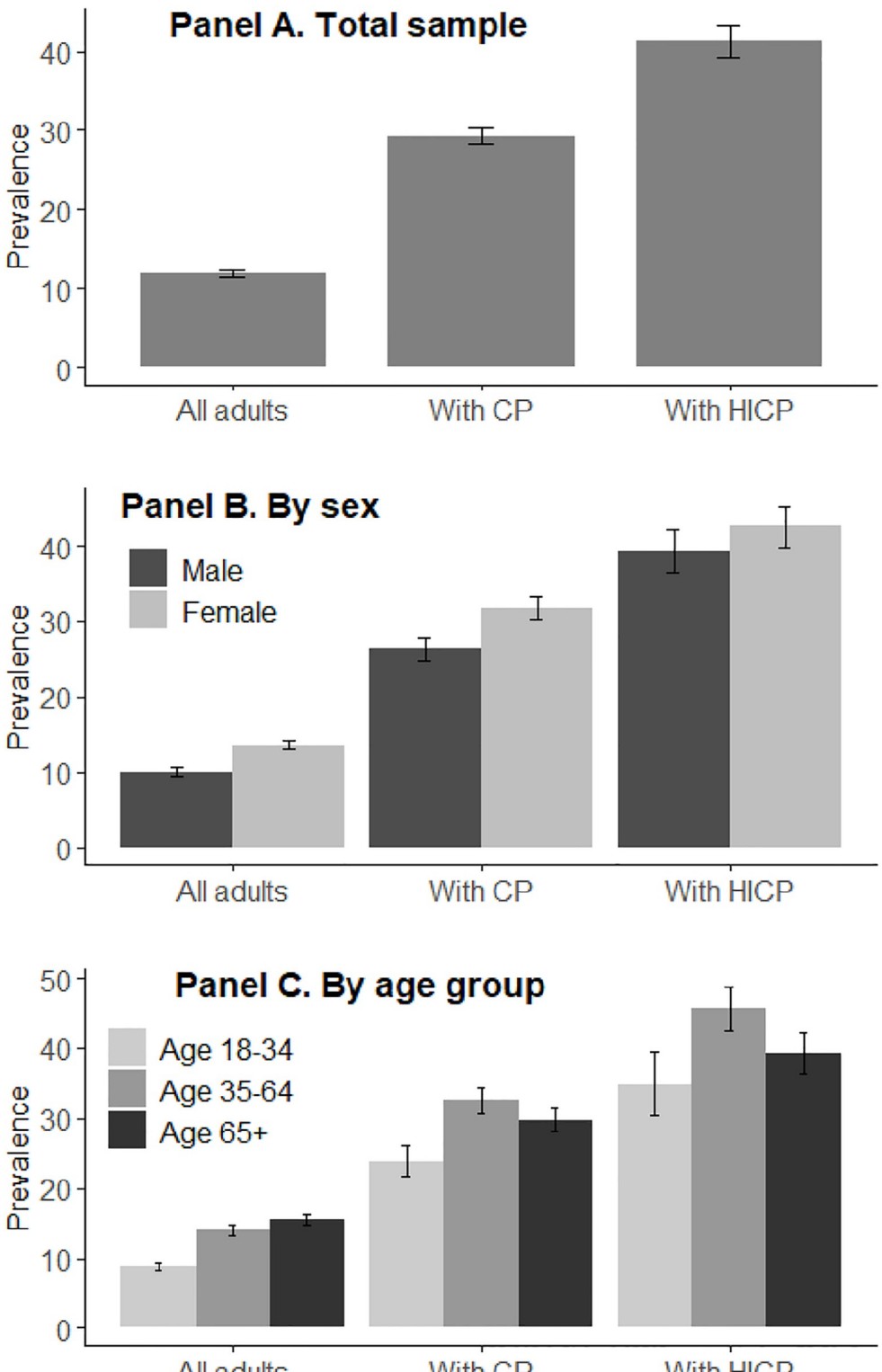

**Fig 1. Prevalence of prior-year prescription opioid use among US adults, by pain status, sex, and age group.**
CP = chronic pain; HICP = high-impact chronic pain. Each plot shows prevalence estimates and their 95% confidence intervals. Estimation adjusts for NHIS complex sampling design. N = 52,617.

**Table 2. Prevalence ratios for POU in fully-adjusted models, US adults 18+, NHIS 2019–2020.**

| | All adults | | With CP | | With HICP | |
|---|---|---|---|---|---|---|
| | PR | 95% CI | PR | 95% CI | PR | 95% CI |
| Year 2020 (ref = 2019) | 0.91** | (0.85,0.96) | 0.87*** | (0.81,0.94) | 0.93 | (0.84,1.02) |
| Region (ref = Northeast) | | | | | | |
| Midwest | 1.28*** | (1.15,1.42) | 1.26*** | (1.11,1.43) | 1.35*** | (1.16,1.58) |
| South | 1.40*** | (1.26,1.55) | 1.36*** | (1.20,1.53) | 1.35*** | (1.16,1.58) |
| West | 1.31*** | (1.17,1.46) | 1.23** | (1.07,1.42) | 1.23* | (1.04,1.45) |
| Rurality (ref = Large central metro) | | | | | | |
| Large fringe metro | 1.01 | (0.92,1.10) | 1.00 | (0.90,1.11) | 0.92 | (0.80,1.06) |
| Medium/small metro | 1.02 | (0.93,1.12) | 0.96 | (0.86,1.07) | 0.97 | (0.86,1.11) |
| Non-metropolitan area | 1.06 | (0.94,1.19) | 0.98 | (0.87,1.11) | 1.01 | (0.86,1.17) |
| Age (ref = 18–44) | | | | | | |
| 45–64 | 1.31*** | (1.21,1.42) | 1.17** | (1.05,1.31) | 1.20* | (1.04,1.38) |
| 65+ | 1.04 | (0.94,1.14) | 0.92 | (0.81,1.05) | 1.00 | (0.86,1.17) |
| Female | 1.23*** | (1.15,1.30) | 1.17*** | (1.08,1.25) | 1.10 | (1.00,1.20) |
| Race (ref = white) | | | | | | |
| Black | 0.92 | (0.83,1.02) | 0.96 | (0.84,1.09) | 0.89 | (0.76,1.04) |
| Hispanic | 0.79*** | (0.71,0.88) | 0.97 | (0.84,1.12) | 1.00 | (0.85,1.17) |
| Other | 0.74*** | (0.62,0.88) | 0.94 | (0.76,1.15) | 0.91 | (0.69,1.21) |
| Immigrant (ref = US-born) | 0.66*** | (0.59,0.74) | 0.83* | (0.71,0.96) | 0.79* | (0.65,0.96) |
| Marital status (ref = married) | | | | | | |
| Previously married | 1.08* | (1.00,1.16) | 1.06 | (0.98,1.15) | 0.93 | (0.83,1.03) |
| Never Married | 0.73*** | (0.67,0.80) | 0.88* | (0.77,1.00) | 0.85 | (0.72,1.00) |
| Education (ref = MA+) | | | | | | |
| Less than high school | 1.16* | (1.01,1.34) | 1.15 | (0.97,1.36) | 1.03 | (0.83,1.29) |
| GED | 1.47*** | (1.24,1.75) | 1.26* | (1.04,1.53) | 1.17 | (0.90,1.51) |
| High school | 1.07 | (0.96,1.18) | 1.05 | (0.91,1.21) | 1.00 | (0.81,1.23) |
| Some college | 1.26*** | (1.14,1.40) | 1.13 | (0.98,1.31) | 1.09 | (0.90,1.33) |
| Associate degree | 1.16** | (1.04,1.29) | 1.12 | (0.97,1.31) | 1.08 | (0.87,1.35) |
| Bachelor's degree | 1.04 | (0.94,1.16) | 0.99 | (0.85,1.16) | 0.94 | (0.75,1.16) |
| Household income (ref = 100k+) | | | | | | |
| $0–34,999 | 1.02 | (0.92,1.13) | 0.94 | (0.82,1.07) | 0.93 | (0.77,1.11) |
| $35,000–49,999 | 1.00 | (0.90,1.12) | 0.95 | (0.83,1.09) | 0.94 | (0.78,1.12) |
| $50,000–74,999 | 1.03 | (0.93,1.14) | 0.97 | (0.85,1.11) | 1.00 | (0.84,1.21) |
| $75,000–99,999 | 0.99 | (0.89,1.10) | 0.92 | (0.79,1.06) | 0.93 | (0.75,1.14) |
| Food insecure (ref = secure) | 1.55*** | (1.43,1.68) | 1.17** | (1.06,1.29) | 1.15** | (1.04,1.29) |
| Not employed (ref = employed) | 1.52*** | (1.42,1.63) | 1.46*** | (1.33,1.61) | 1.17* | (1.03,1.33) |
| Health insurance (ref = private) | | | | | | |
| No insurance | 0.66*** | (0.57,0.77) | 0.65*** | (0.53,0.80) | 0.54*** | (0.40,0.73) |
| Public insurance | 1.19*** | (1.11,1.27) | 1.03 | (0.96,1.12) | 0.94 | (0.85,1.03) |
| N | 50,773 | | 11,773 | | 4,298 | |

* p < .05,

** p < .01,

*** p < .001.

Findings statistically significant at .05 or less are shaded in gray.

POU = prescription opioid use; CP = chronic pain; HICP = high-impact chronic pain; PR = prevalence ratio; CI = confidence interval; ref = reference category;

GED = general educational development diploma; MA = master's degree.

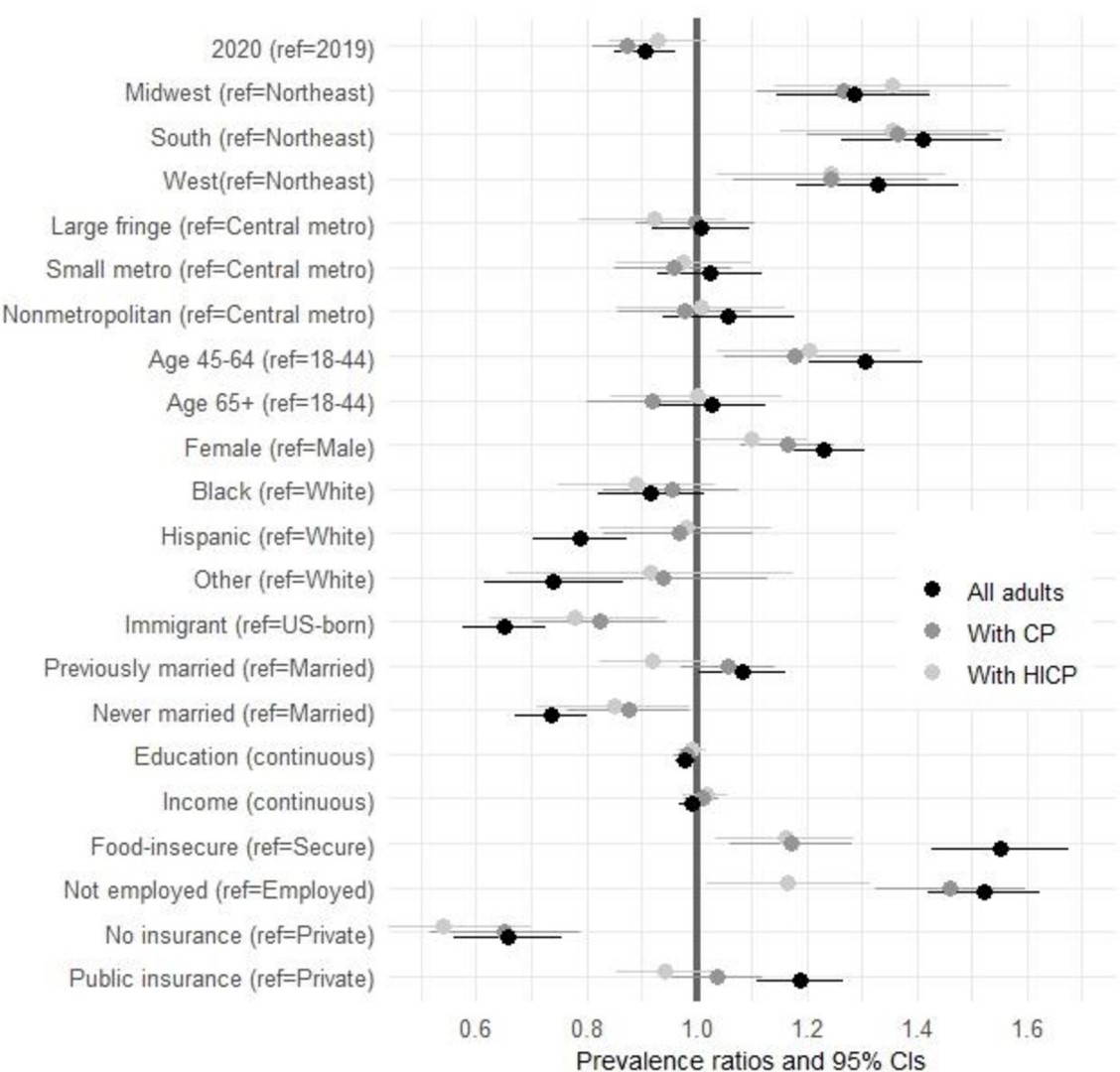

**Fig 2. Coefficient plot showing the association between all covariates and prescription opioid use in fully-adjusted models, for three subsamples: All adults, adults with chronic pain, and adults with high-impact chronic pain.** The plot shows prevalence ratios and their 95% confidence intervals from fully-adjusted robust Poisson models of prescription opioid use estimated separately for the three subsamples of all adults, adults with CP, and adults with HICP. The omitted category for each categorical predictor is in parentheses. CP = chronic pain; HICP = high-impact chronic pain; metro = metropolitan area.

findings were comparable in both years. Additional sensitivity analyses used continuous age and age squared, with results similar to those shown below with categorical age.

## Results

NHIS interviewed 31,997 adults in 2019 and additional 21,153 new respondents in 2020, for a total of 53,150 respondents. Of this total, 533 respondents (1%) were missing information on POU, yielding an effective analytic sample size of 52,617 for unadjusted POU prevalence estimates. S1 Table in S1 File shows the number of missing values for each covariate. The missingness ranges from 0% (e.g., geographic characteristics) to below 3% (e.g., immigrant status, marital status). Altogether, among those with valid POU information, 1,844 respondents

(3.5%) were missing one or more predictors; thus, the sample size for the total subsample, which include all covariates, is 50,773.

The two analytic subsamples of adults with pain comprise the 12,124 adults who reported they had chronic pain (CP) and 4,447 who reported high-impact chronic pain (HICP). These are the sample sizes for POU prevalence estimates in the corresponding subgroups, with samples for regression models about 3% smaller due to missingness on predictors. The prevalence of CP and HICP, 21.2% and 8.2%, respectively, is similar to published estimates [21,23,31].

## Prevalence of POU

Among US adults, as Table 1 summarizes and Fig 1 depicts graphically, 11.9% (95% CI 11.5, 12.3) report having used prescription opioids during the past 12 months. The prevalence is 29.3% (95% CI 28.2, 30.4) among those with CP and 41.2% (95% CI 39.2, 43.2) with HICP. Women have higher POU than men, about 3.5 percentage points higher in the general population, although among those with HICP the confidence intervals overlap. Among all adults, the age pattern is monotonic, with older adults using more opioids. However, among those with CP and HICP, the pattern is nonlinear: middle-aged Americans with pain have the highest POU. Indeed, nearly half (45.7%, 95% CI 42.7, 48.8) of US adults age 45–64 with HICP report prior-year POU.

## Association of POU with time, geographic, demographic, and socioeconomic factors

**Time 'trend'.**   As Table 2 and Fig 2 show, the prevalence of POU is about 9% lower in 2020 compared with 2019, net of all other covariates, in the general population (PR = 0.91, 95% CI 0.85, 0.96), as well as among those with CP; the decline is not statistically significant in the subgroup of adults with HICP although the point estimate is also in the downward direction.

Geographic factors show large differences by region, but no differences by rurality. The regional differences are statistically significant in all three subgroups and substantively large: In the general population, POU prevalence is 28% higher in the Midwest, 31% higher in the West, and 40% higher in the South than in the Northeast; the respective prevalence ratios among adults with CP are 26%, 23%, and 36% higher, all statistically significant These are large differences in absolute terms. As S2 Table in S1 File shows, these regional differences translate to, for example, 11.6 percentage point higher POU in the South compared with Northeast among those with HICP, net of all other covariates.

In contrast, there is no POU difference across the rural/urban continuum in any of the three analytic samples. Robustness checks (not shown) find no rural/urban difference in POU in any of the four census regions, either.

**Demographic characteristics.**   Taking into account the distribution of other covariates, POU is highest for Americans 45–64, in all three analytic samples, and statistically comparable among those 18–44 and 65+. Women in the general population report higher POU than men (PR = 1.23, 95% CI 1.15, 1.30); this is due to the higher pain burden among women: The gender POU difference is slightly smaller among adults with CP, with women's POU 17% higher, and 'only' 10% among adults with HICP (not statistically significant). Race/ethnic differences are complex: there are no significant white/Black POU differences net of all other covariates in either subsample. Hispanic and "other" adults in the general population report lower POU than whites, due to unequal pain burden: among those with CP or HICP, there are no statistically significant race/ethnic differences. In contrast, immigrants have 34% lower POU rates

than the US-born (PR = 0.66, 95% CI 0.59, 0.74), a difference attenuated but still substantively large and statistically among those with CP and HICP.

**Socioeconomic and insurance status.** Food-insecure and non-employed adults have dramatically higher POU than the food-secure and employed, respectively, net of all covariates. The differences are substantively large and statistically significant in all adults, although it's attenuated among those with CP and HICP. Food-insecure Americans report 55% higher prevalence of POU compared with food-secure adults (PR = 1.55, 95% CI 1.43, 1.68). Much of this excess is due to their high pain burden though: among adults with CP and HICP, opioid use is 'only' 17% and 15% higher, respectively, compared with their food-secure counterparts, as Fig 2 highlights. Not being employed is also associated with much higher POU than being employed (PR = 1.52, 95% CI 1.42, 1.63). This difference isn't due to any level of chronic pain but only HICP where POU is 17% higher among the not employed.

In contrast, net of all characteristics, income is not associated with POU in any of the three analytic subgroups. The role of education is also somewhat complex and nonlinear. In the general population, adults with a high school diploma have comparable POU as their college-educated counterparts, while all others with less than high school but also those with postsecondary schooling short of a bachelor's degree have significantly higher POU. In any case, but this pattern appears entirely due to underlying differences in pain burden: among adults with CP and especially HICP, there are no educational differences in POU. Finally, adults without health insurance are dramatically less likely to take prescription opioids compared with adults who have private insurance, whether in the full sample (34% lower prevalence) or among those with CP (35% lower) and especially HICP (46% lower).

Overall, thus, POU prevalence is highest in the South and among those who are middle-aged, food insecure, and not employed. POU prevalence is lowest in the Northeast and among immigrants and adults without health insurance.

## Discussion

Recent research has documented changes in prescription opioid use (POU) among US adults over the past decade: prescription rates have declined overall [32,33], even as some dimensions of use such as duration may have increased [34]. However, little is known about current national POU patterns across region and rurality, sociodemographic characteristics, and pain status. Using large, nationally representative data collected in 2019–2020, we present important new results regarding POU levels and disparities in the US adult population, as well as specifically among Americans with pain.

Overall, we find high prevalence of POU: 11.9% of US adults reported having used opioids in the past 12 months, which translates to 30.7 million Americans. POU prevalence is higher among those with pain: 29.3% for adults with chronic pain (CP) and 41.2% for those with high-impact chronic pain (HICP). The roughly 12% prevalence we identified for the general population is similar to the prior-year POU prevalence of 12–14% estimated from the Medical Expenditure Panel Survey [7,35], and similar to prevalence estimates of 12–13% from Canada, the country with the second-highest per-capita opioid consumption after the US [36–38]. Possibly due to different usage time frames, our estimates are higher than those for prior-month use estimated from NHANES, which hover around 6.5% [6,39], or for current use (8% among older adults in the Health and Retirement Study) [5]. Our estimates are considerably lower than prior-year prevalence estimates from the National Survey of Drug Use and Health, which exceed 30% [40,41]. Differences in question wording including but not restricted to the time horizon, as well as other differences across surveys, likely explain this variability in estimates.

Our estimate for POU prevalence among respondents with chronic pain (29.3%) is slightly higher than the 27% found in a meta-analysis from 60 available studies worldwide using data from various years between late 1990s and 2010s [42], and much higher than estimates from individual countries outside the US [43–48]. This suggests that prescription opioid use rates among chronic pain patients remain higher in the US than in many other countries.

Overall, our estimates indicate continued heavy reliance on prescription opioids, and provide important data points for the ongoing national efforts to change the prescription patterns [49,50]. We note, however, that POU prevalence was about 10% lower in 2020 than in 2019, possibly continuing the decline observed over the prior decade [49], although perhaps also due to COVID-related disruptions to health care.

The data revealed unexpected geographic patterns in POU: large regional disparities, but no rural/urban differentials. Adults in the South have 41% higher POU prevalence, and in the Midwest and West about 30% higher prevalence, than their peers in the Northeast. Importantly, these estimates account for demographic compositional differences (in age, race/ethnicity, sex, etc.) across regions. Moreover, the regional differences in POU are not primarily attributable to differences in pain prevalence: the same regional POU patterns, attenuated only somewhat, are found among adults with pain. These findings corroborate scattered prior reports finding prescription opioid use to be highest in the Southeast, Appalachian, or South Central regions [32,51].

In contrast, POU does not vary across the rural/urban continuum, either in the general population or among adults with pain. This finding is surprising given widespread perception of higher opioid use in rural areas, supported by studies reporting large rural/urban differences in opioid misuse and related morbidity and mortality [52]. Specifically for *prescription* opioid use, however, the sparse prior research yields conflicting findings: Some studies find higher POU in rural areas [13,15], others find lower [9], and still others find no rural/urban differences [5,12,53], as in the current study. Many accounts of the opioid epidemic focus on opioid misuse in rural areas, but our findings suggest that large and medium urban areas are not necessarily less vulnerable, at least not based on POU rates.

We also highlight important demographic correlates of POU. Net of other covariates, women have about 23% higher POU prevalence than men, a finding corroborating prior reports [7,35]. However, our analysis shows that a large part of this difference is due to women's greater pain burden: indeed, among adults with HICP, the gender difference in POU is not statistically significant.

Racial/ethnic patterns suggest relatively modest differences net of other covariates. We find lower POU among Hispanic and 'other' adults in the general population, but not among adults with pain, and there are no Black-white differences in any analytic subsample. These results clash with the widespread understanding of pronounced racial disparities in opioid treatment [54], but they echo the most current results from the Medical Expenditure Panel Survey [35], and approximate bivariate estimates from the 2019 NHIS for the general population [12], which also found modest Black-white differences. This raises the possibility that changes in opioid prescribing over the past decade have decreased racial disparities in POU and should be examined in further studies.

In contrast, differences by immigrant status are substantial. Immigrants have considerably lower POU than the US-born: 34% lower among all adults and 17% to 21% lower among those with CP and HICP, respectively, generally corroborating prior reports [16,35]. Whether this difference protects immigrants from overreliance on opioids or instead indicates a particularly pronounced undertreatment of pain—a very important question relevant to all disparities described herein—is unfortunately beyond the scope of our analysis.

With respect to socioeconomic status (SES), prior work has consistently found higher POU among lower-SES adults [5,9,53,55]. What our analysis contributes is that food insecurity and employment status are more salient predictors of POU use than the more commonly used SES measures of education or income, which were largely not significant net of these other covariates. Food-insecure adults have 55% higher prevalence of POU, and non-employed adults have 52% higher prevalence, than their food-secure and employed counterparts, even net of all other covariates. These disparities remain significant albeit smaller even among those with pain. The strong relationship between food insecurity and POU was recently described for Canada [56], and a US report also highlighted POU disparities by employment status [6,12]. Such findings, identifying dimensions of SES that are particularly salient for pain and pain treatment disparities, have a clear public health policy relevance.

Finally, relatively little prior work has addressed POU differences by health-insurance status. We find some of the insurance-related differences to be quite large. Compared to adults with private insurance, uninsured adults in the general population have 34% lower prevalence of POU; moreover, the gap is greater still among those with pain–and thus most likely in need of opioids—uninsured adults with HICP have 46% lower use than those with private insurance. While at first glance this may seem inconsistent with findings that the proportion uninsured at the *county level* is linked with greater use [9], as we noted in the introduction, that such county-level variables capture a county's average SES rather than access to health care for any given individual. Our study also finds differences in POU between those with public and private insurance, but only in the general population, where adults with public insurance like Medicaid have 19% higher prevalence of POU. This difference ceases to be significant when we restrict our sample to those with pain. That is, among adults with pain, public vs. private insurance makes no difference in POU. A recent study reported higher POU among adults with public versus private insurance in simple bivariate association [12], but did not adjust for age, SES, pain, etc., preventing further comparison to our study. In sum, not having affordable access to health care due to lack of insurance is powerfully associated with POU, whereas type of insurance (public vs. private) does not appear to affect access to prescription opioids net of pain status.

Our study has a number of strengths compared to previous research. First, by conducting analyses separately for all adults, adults with CP, and adults with HICP, we clarify to what extent POU differences in the general population reflect disparities in pain burden. For instance, in the general population, opioid use prevalence is 55% higher among food-insecure Americans compared with their food-secure counterparts, net of all other characteristics. Among those with pain, however, the disparity is 'only' 15–17%. Our findings thus demonstrate that higher opioid use among food-insecure Americans is to a large extent—although not entirely—linked to their higher pain burden. Similar assessments can be done for all other covariates included in our analysis. Additional strengths stem from our use of the NHIS data. NHIS, the leading national survey on US adults' health, provides complex survey adjustment to represent the US non-institutionalized population. The data comprise a rich array of relevant covariates collected at the individual level. This is a major advantage over studies based on opioid dispensing information at the county or state levels that are subject to ecological fallacy, as well as over Medicare claims data that only include basic demographic information.

At the same time, other factors limit the generalizability and impact of our findings. First, the NHIS response rate, while respectable (59.1% in 2019 and 48.9% in 2020), is lower than ideal, and could conceivably introduce some selection tendencies impacting our estimates. Second, certain variables provide sub-optimal levels of detail. NHIS's geographic regions, for example, are too large to support specific policy recommendations. Future work, when more waves of data accumulate, should use the restricted NHIS data to conduct state-specific

analyses [57,58]. Regarding POU, it would be helpful to distinguish between weak opioids such as codeine, strong opioids such as oxycodone or fentanyl, and opioids used to treat opioid use disorder, such as methadone or buprenorphine. It would also be important to better characterize exposure to opioids, e.g., how long respondents have taken opioids, for what conditions, and at what dosages. Information about opioid misuse would also be desirable, although it is unlikely that such information could be collected in a general health survey like the NHIS. One specific item would be a feasible and an important addition: the NHIS has previously asked about prior-year surgery; returning this item to the survey could help distinguish post-surgical POU from other categories of POU. Finally, a weakness of the NHIS is that opioid use is self-reported and thus may be reported inaccurately, especially over the relatively long 12-month time interval.

## Conclusions

Our study finds that nearly a third of US adults with chronic pain, and over 40% with high-impact chronic pain, have taken prescription opioids in the past year. With respect to geography, we found no rural-urban differences in opioid use. In contrast, the striking geographic disparities across Census regions, with highest use in the South even net of individual-level characteristics and pain status, are particularly important and relevant for policy, as they may indicate systematic approaches to treatment of pain that could be effectively targeted by federal or state-level policy interventions. Aspects of disadvantaged social status do not all affect POU in the same way. Those that limit access to health care, such as lack of insurance and possibly also immigrant status, reduce POU, while those that may be viewed as reflecting financial stress, such as food insecurity or unemployment, increase POU. This tension underscores the need to analyze socioeconomic predictors of POU in fine-grained detail, taking into account the interplay of competing effects of multiple socioeconomic factors in shaping opioid use. Our study cannot answer whether POU prevalence is too high or too low, but our findings contribute to the evidence base for policymakers and practitioners addressing this important question. Documenting prescription opioid use levels and patterns is an essential step in minimizing disparities in POU, guiding opioid risk mitigation strategies, and improving prevention and management of both pain and addiction for all Americans.

## Supporting information

**S1 File.**
(DOCX)

## Acknowledgments

The authors thank Dr Zachary Zimmer, Dr Feinuo Sun, and Harry Alorgbey Sardina for their assistance, suggestions, and comments on previous drafts of this article.

The content is solely the responsibility of the authors and does not necessarily represent the official views of the National Institutes of Health or SSHRC.

## Author Contributions

**Conceptualization:** Anna Zajacova, Hanna Grol-Prokopczyk, Ian Gilron.

**Formal analysis:** Anna Zajacova, Christopher Schwarz.

**Funding acquisition:** Anna Zajacova, Hanna Grol-Prokopczyk.

**Investigation:** Merita Limani.

**Methodology:** Christopher Schwarz.

**Project administration:** Merita Limani.

**Visualization:** Christopher Schwarz.

**Writing – original draft:** Anna Zajacova, Hanna Grol-Prokopczyk, Ian Gilron.

**Writing – review & editing:** Anna Zajacova, Hanna Grol-Prokopczyk, Merita Limani, Ian Gilron.

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
