## [Decision Letter · Decision Letter 0]

20 Dec 2022

PONE-D-22-25884Prevalence and Correlates of Prescription Opioid Use among US Adults in 2019-2020PLOS ONE

Dear Dr. Zajacova,

Thank you for submitting your manuscript to PLOS ONE. After careful consideration, we feel that it has merit but does not fully meet PLOS ONE’s publication criteria as it currently stands. Therefore, we invite you to submit a revised version of the manuscript that addresses the points raised during the review process. Both reviewers found considerable merit with the current version of the manuscript; however, they also asked for some minor clarifications. Please do your best to respond to the reviewers' comments.

We look forward to receiving your revised manuscript.

Kind regards,

Kenzie Latham-Mintus, PhD, FGSA

Academic Editor

PLOS ONE

Journal Requirements:

"Research reported in this analysis was supported by the National Institute on Aging of the National Institutes of Health under Award Number R01AG065351 (PI: Grol-Prokopczyk) and

by the Social Sciences and Humanities Research Council of Canada (SSHRC) Insight Grant (PI: Zajacova). The content is solely the responsibility of the authors and does not necessarily represent the official views of the National Institutes of Health or SSHRC"

"Research reported in this analysis was supported by the National Institute on Aging of the National Institutes of Health under Award Number R01AG065351 (PI: Grol-Prokopczyk) and

by the Social Sciences and Humanities Research Council of Canada (SSHRC) Insight Grant (PI: Zajacova). The content is solely the responsibility of the authors and does not necessarily represent the official views of the National Institutes of Health or SSHRC.

Additional Editor Comments (if provided):

Reviewers' comments:

Reviewer's Responses to Questions

**Comments to the Author**

1. Is the manuscript technically sound, and do the data support the conclusions?

Reviewer #1: Yes

Reviewer #2: Yes

2. Has the statistical analysis been performed appropriately and rigorously? 

Reviewer #1: Yes

Reviewer #2: Yes

3. Have the authors made all data underlying the findings in their manuscript fully available?

Reviewer #1: Yes

Reviewer #2: Yes

4. Is the manuscript presented in an intelligible fashion and written in standard English?

Reviewer #1: Yes

Reviewer #2: Yes

5. Review Comments to the Author

Reviewer #1: This is an interesting paper which describes the prevalence of opioid use in the USA and factors associated to opioid use. The study is well conducted and its results well presented, although some improvements in Methods are necessary.

Abstract

Please define the acronyms at their first appearance and use them through the abstract.

Insert some more quantitative results.

Introduction

Please define the acronyms at their first appearance and use them through the text.

Please delete the first sentence “This analysis presents a comprehensive portrait of prescription opioid use (POU) among US adults in 2019/2020.”

Methods

Please move the first sentence at the end of the Data section.

Please include the sources mentioned in the sentences “are available 78 here for 2019 and here for 2020.” and “Full 87 details about the prompt are available on page 287 here.” among the References.

Please move the sentence “All variables are self-reported” at the end of the Variables section.

Please remove the variable names in the Variables section.

Please delete this sentence “We coded the outcome as 1 if 88 respondents reported taking prescription opioids in the past 12 months and 0 otherwise.”

Please explain what the authors mean by food insecurity.

Please rename the section Approach into Statistical analyses.

The paragraph from line 114 to line 124 should be moved in the Results section.

Please add a few more details on the Statistical Methods.

Please add the source “NHIS IPUMS guidelines” in the References.

Figure 2

Please check the data for food-insecure vs secure. The results for all individuals are very different from those of the two subgroups.

Table 2 and Supplementary Tables 1 and 2

Please provide the acronym definition in the footnotes.

Reviewer #2: The article is well written and acknowledges the problems with this type of database--self-reporting for example.

Please define the term "pain patient" on line 51, especially in light of the later classifications of individuals with pain and individuals with high impact pain.

6. PLOS authors have the option to publish the peer review history of their article (what does this mean?). If published, this will include your full peer review and any attached files.

Reviewer #1: No

Reviewer #2: No

---

## [Decision Letter · Decision Letter 1]

17 Feb 2023

Prevalence and Correlates of Prescription Opioid Use among US Adults, 2019-2020.

PONE-D-22-25884R1

Dear Dr. Zajacova,

We’re pleased to inform you that your manuscript has been judged scientifically suitable for publication and will be formally accepted for publication once it meets all outstanding technical requirements.

Kind regards,

Kenzie Latham-Mintus, PhD, FGSA

Academic Editor

PLOS ONE

Additional Editor Comments (optional):

Reviewers' comments:

Reviewer's Responses to Questions

**Comments to the Author**

1. If the authors have adequately addressed your comments raised in a previous round of review and you feel that this manuscript is now acceptable for publication, you may indicate that here to bypass the “Comments to the Author” section, enter your conflict of interest statement in the “Confidential to Editor” section, and submit your "Accept" recommendation.

Reviewer #1: All comments have been addressed

Reviewer #2: All comments have been addressed

2. Is the manuscript technically sound, and do the data support the conclusions?

Reviewer #1: Yes

Reviewer #2: Yes

3. Has the statistical analysis been performed appropriately and rigorously? 

Reviewer #1: Yes

Reviewer #2: Yes

4. Have the authors made all data underlying the findings in their manuscript fully available?

Reviewer #1: Yes

Reviewer #2: Yes

5. Is the manuscript presented in an intelligible fashion and written in standard English?

Reviewer #1: Yes

Reviewer #2: Yes

6. Review Comments to the Author

Reviewer #1: The authors have properly addressed all the comments. The paper can now be accepted for publication.

Reviewer #2: All concerns have been addressed. The authors have revised the manuscript extensively in response to reviewer #1 comments. The findings remain as before but are better described and the language is more rigorous and more clear. The ambiguities in use of acronyms and other terms have been addressed.

7. PLOS authors have the option to publish the peer review history of their article (what does this mean?). If published, this will include your full peer review and any attached files.

Reviewer #1: **Yes: **Cristina Bosetti

Reviewer #2: No

---

## [Editor Report · Acceptance letter]

22 Feb 2023

PONE-D-22-25884R1 

Prevalence and Correlates of Prescription Opioid Use among US Adults, 2019-2020. 

Dear Dr. Zajacova:

I'm pleased to inform you that your manuscript has been deemed suitable for publication in PLOS ONE. Congratulations! Your manuscript is now with our production department. 

Kind regards, 

on behalf of

Dr. Kenzie Latham-Mintus 

Academic Editor

PLOS ONE